# Fine-scale distribution of malaria mosquitoes biting or resting outside human dwellings in three low-altitude Tanzanian villages

**Arnold S. Mmbando** [1]*, **Emmanuel W. Kaindoa**[1,2], **Halfan S. Ngowo**[1,3], **Johnson K. Swai**[1], **Nancy S. Matowo**[1,4,5], **Masoud Kilalangongono**[1], **Godfrey P. Lingamba**[1†], **Joseph P. Mgando**[1], **Isaac H. Namango**[1,4,5], **Fredros O. Okumu**[1,2,3,6☯], **Luca Nelli**[3☯]

1 Environmental Health and Ecological Sciences, Ifakara Health Institute, Ifakara, Tanzania, 2 Faculty of Health Sciences, School of Public Health, University of the Witwatersrand, Parktown, Republic of South Africa, 3 Institute of Biodiversity, Animal Health and Comparative Medicine, University of Glasgow, Glasgow, United Kingdom, 4 Swiss Tropical and Public Health Institute, Basel, Switzerland, 5 University of Basel, Basel, Switzerland, 6 School of Life Science and Bioengineering, Nelson Mandela African Institution of Science & Technology, Arusha, Tanzania

☯ These authors contributed equally to this work.
† Deceased.
* ammbando@ihi.or.tz

**Data Availability Statement:** All relevant data are within the manuscript and its Supporting Information files.

## Abstract

### Background

While malaria transmission in Africa still happens primarily inside houses, there is a substantial proportion of *Anopheles* mosquitoes that bite or rest outdoors. This situation may compromise the performance of indoor insecticidal interventions such as insecticide-treated nets (ITNs). This study investigated the distribution of malaria mosquitoes biting or resting outside dwellings in three low-altitude villages in south-eastern Tanzania. The likelihood of malaria infections outdoors was also assessed.

### Methods

Nightly trapping was done outdoors for 12 months to collect resting mosquitoes (using resting bucket traps) and host-seeking mosquitoes (using odour-baited Suna® traps). The mosquitoes were sorted by species and physiological states. Pooled samples of *Anopheles* were tested to estimate proportions infected with *Plasmodium falciparum* parasites, estimate proportions carrying human blood as opposed to other vertebrate blood and identify sibling species in the *Anopheles gambiae* complex and *An. funestus* group. Environmental and anthropogenic factors were observed and recorded within 100 meters from each trapping positions. Generalised additive models were used to investigate relationships between these variables and vector densities, produce predictive maps of expected abundance and compare outcomes within and between villages.

### Results

A high degree of fine-scale heterogeneity in *Anopheles* densities was observed between and within villages. Water bodies covered with vegetation were associated with 22% higher

**Funding:** Fredros Okumu was funded by the Wellcome Trust Intermediate Research Fellowship (Grant number: WT102350/Z/13/Z) which funded this research. Arnold Mmbando was also supported by the Wellcome Trust Masters Fellowship in Public Health and the Association of Physicians of Great Britain and Ireland for funding this research (Grant number 106356/Z/14/Z).

**Competing interests:** The authors declare that they have no competing interests.

densities of *An. arabiensis* and 51% lower densities of *An. funestus*. Increasing densities of houses and people outdoors were both associated with reduced densities of *An. arabiensis* and *An. funestus*. Vector densities were highest around the end of the rainy season and beginning of the dry seasons. More than half (14) 58.3% of blood-fed *An. arabiensis* had bovine blood, (6) 25% had human blood. None of the *Anopheles* mosquitoes caught outdoors was found infected with malaria parasites.

## Conclusion

Outdoor densities of both host-seeking and resting *Anopheles* mosquitoes had significant heterogeneities between and within villages, and were influenced by multiple environmental and anthropogenic factors. Despite the high *Anopheles* densities outside dwellings, the substantial proportion of non-human blood-meals and absence of malaria-infected mosquitoes after 12 months of nightly trapping suggests very low-levels of outdoor malaria transmission in these villages.

## Background

Recent advances in malaria control are mostly attributed to scale-up of preventative and treatment measures including, long-lasting insecticidal bed nets (LLINs), indoor residual spraying (IRS), rapid diagnostic tests (RDTs), and artemisinin-based combination therapy (ACT) [1]. In Tanzania, the residual burden varies between districts, but approximately 60% of people are still living in areas having moderate or high burden [2]. As in many other settings, malaria vector control in Tanzania faces several challenges, notably mosquito resistance to commonly used insecticides, and changes in vector behaviours that result in avoidance of indoor control tools [3]. *Anopheles* mosquitoes that bite or rest outdoors are not readily tackled by LLINs or IRS, and therefore can perpetuate residual disease transmission [4]. Moreover, LLINs and IRS may themselves exacerbate outdoor-biting and resting, thereby worsening outdoor malaria exposure [5,6].

Outdoor biting has been reported in major African malaria vectors, namely *Anopheles arabiensis* and *An. funestus*, but also in secondary vector species such as *An. rivulorum*, *An. coustani* and *An. ziemanni*, which can become important vectors in areas where LLINs and IRS effectively controlled *An. gambiae* and *An. funestus* [7,8]. Also, human behaviours have been linked with the persistent malaria transmission [9]. Thus, additional mosquito control tools are highly needed to control residual malaria vectors in order to drive transmission to zero [10,11].

While there have been several studies on malaria mosquitoes caught indoors, there is insufficient data on population densities, behaviours and transmission activity of outdoor-biting or outdoor-resting populations. This is partly due to insufficient or lack of appropriate sampling techniques [12,13]. Malaria risk is traditionally mapped using models based on remotely-sensed imagery of climatic and environmental variables to determine spatial and temporal patterns of disease risks at broad-scale [14,15]. However, predictive maps based on such coarse covariates typically lack fine-scale explanatory power for local decision making [16]. Besides, heterogeneities of malaria transmission, especially in low-transmission settings, hinders accurate prediction of residual malaria transmission risks even when risk maps optimized with geo-information techniques [17]. Thus, most malaria risk maps have a poor predictive capacity

at a community level and need additional surveillance of malariometric measures from the specific time and place [18].

Improved ecological models for transmission risk can improve outcomes [16], especially if local disease data is incorporated in simple interactive models [19]. Through mosquito surveillance systems, we can identify places with a high risk of residual malaria transmission (hot spot) and places with low transmission (cold spot). Variations of resistant malaria vector population densities could be predicted in terms of space (spatial) and time (temporal), which will help to identify the correct seasons and locations to apply the complementary tools.

To accelerate malaria control towards eventual elimination at local level, there is need for a careful integration of both large-scale and fine-scale surveillance systems to account for variations of risk, associated with mosquitoes resistant to insecticides, or those biting outdoors [16]. This current study investigated the distribution of malaria mosquitoes biting or resting outside dwellings in three low-altitude Tanzanian villages where LLINs are highly used and malaria transmission persists. The study villages are characterized by high pyrethroid-resistant *Anopheles* mosquitoes. The study focused on outdoor mosquito populations as opposed to indoor populations, and also assessed the likelihood of malaria infections in these villages.

## Methods

### Study area

This study was conducted in three low-altitude wards (Kivukoni; 8.2135$^o$S & 36.6879$^o$E, Minepa; 8.2710$^o$S & 36.6771$^o$E, and Mavimba: 8.3124$^o$S & 36.6771$^o$E) in the malaria-endemic Kilombero valley in South-Eastern Tanzania (Fig 1). Data collection was done between February 2015 and February 2016. The main economic activity in the area is rice cultivation, but other land use activities included fishing, forestry and livestock-keeping [20]. Annual rainfall is 1200–1800 mm, with intermittent rainfall from November to January, followed by more consistent rainfall starting from March to May. Mean daily temperatures are 20˚C - 33˚C, and altitude ranges between 120 to 350 m above sea level [21,22]. Primary malaria vectors are *Anopheles funestus* and *Anopheles arabiensis*, both of which are resistant to pyrethroids [5,23,24]. Other *Anopheles* spp. e.g. *An. coustani*, *An. pharoensis*, *An. wellcomei* and *An. ziemmani*, are also present together with non-malaria mosquitoes such as *Mansonia*, *Culex* and *Aedes* species.

### Selection of outdoor mosquito sampling units

Square grids (200m × 200m) were overlaid on the village maps, and the centroids (grid points) of each cell was geo-referenced as described by Mwangungulu *et al* [21]. A total of 270 grid points with human settlements were randomly selected for mosquito sampling across the three villages: 118 in Kivukoni, 86 in Mavimba and 66 in Minepa. Two sentinel grid points were assigned in each village. Mosquitoes were sampled from each of these sentinel grid points for ten nights in each round (totalling 30 trap-nights/month), repeated ten times from February 2015 to February 2016.

In addition to the sentinel grid points, four others were randomly selected to sample mosquitoes each night. The randomly sampled grids remained were used for nightly repeat sampling for a month, before another set of four random grid points was selected. Following this sampling design, each grid point in each village was visited once per experimental round for the 10 rounds. Therefore, each village had six grids being sampled in each night (four that were randomly selected every month and two which remained fixed). The fixed grids enabled assessment of both spatial and temporal patterns, while the randomly-selected grids enabled assessment of only spatial distribution of malaria vectors.

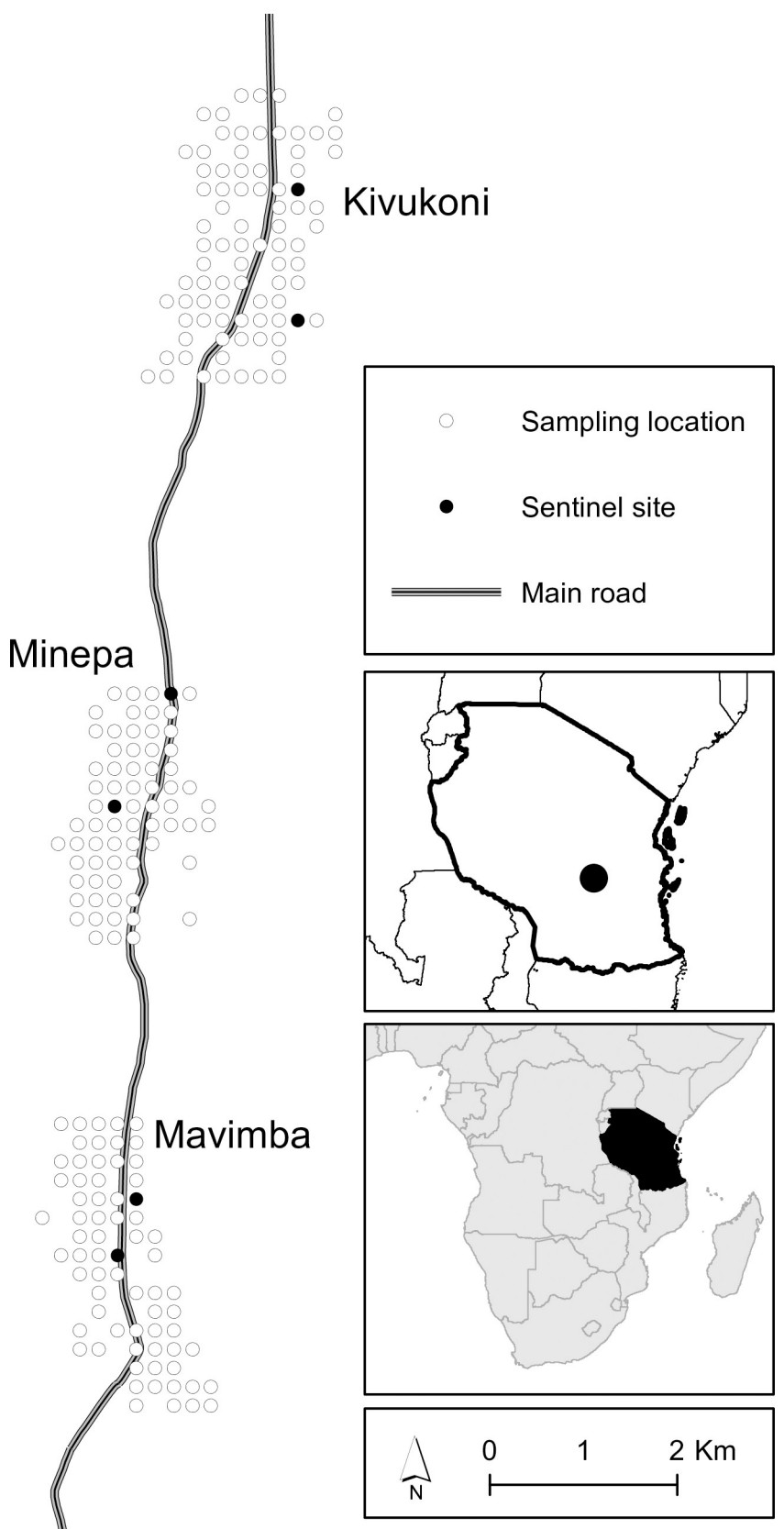

Fig 1. **Location of the study area in Tanzania and sampling points (administrative borders source: Data.humdata.org.**

## Mosquito traps and attractant

Mosquitoes host-seeking outdoors were trapped using odour-baited Suna® Traps, while those resting outdoors were trapped using resting Bucket Traps (RBu) (Fig 2). The Suna® traps were suspended 30 cm above ground like in the previous studies (Fig 2A) [25]. The Suna® traps proved to catch significantly higher number of *Anopheles* species in the field conditions as well as it significantly reduce entry of malaria mosquitoes inside the experimental huts [25,26]. The traps were baited with synthetic attractants dispensed in pellets and supplemented with carbon dioxide gas from yeast-molasses fermentation [27–29]. Yeast-molasses mixtures (35gm yeast,500ml molasses, and 2L water) were prepared one hour before starting mosquito collections each night [30].

The RBu traps on the other hand, consisted of 20 litre plastic buckets lined with black cotton lining on the inside to provide dark environments for mosquitoes to hide and rest (Fig 2B). The traps were set at dusk and a small wet cloth placed inside to provide humid condition which favoured mosquito resting [31]. The mosquitoes found resting inside the RBu traps were aspirated out early each morning [32].

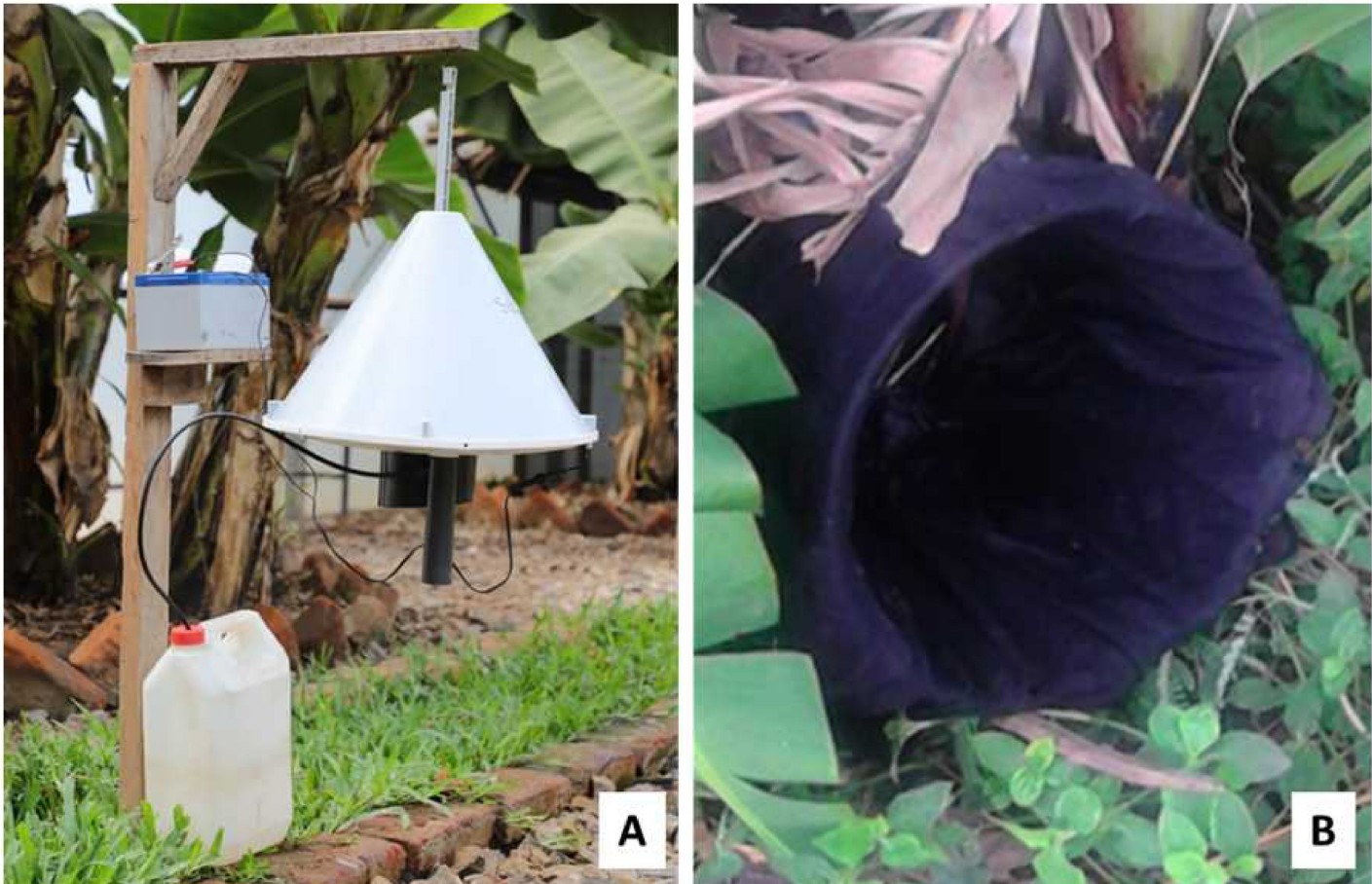

Fig 2.  Mosquito Traps: Odour-baited Suna® trap, for sampling host-seeking mosquitoes (A) and Resting bucket trap (RBu) for resting mosquitoes (B).

Each sampling night, the Suna® traps were placed at the centroid of the selected grids, while the RBu traps were situated 50m away from the Suna® traps. Mosquitoes sampling was done nightly between 6pm and 7am each night.

## Assessment of environmental and anthropogenic conditions in the sampling areas

For each grid point, environmental and anthropogenic features potentially affecting outdoor mosquito densities were assessed and recorded within 100m radius from each trap location. First, land use activities like mining, crop-cultivation and house construction events were observed and recorded. Second, land cover characteristics were identified, namely open land, grassland, wetland, seasonal swamp, river stream, shrub and forest. Third, distances to the nearest household, road, cowshed, toilet and water-bodies were also recorded, by using the hand-held GPS. Lastly, anthropogenic activities such as time when people went indoors each night, presence and densities of domestic animals and poultry in peri-domestic areas, and occurrence of leisure activities were observed and recorded. If a certain factor was present in a certain map grid was recorded by "yes" otherwise "no".

## Mosquito identification

All mosquitoes collected were retrieved each morning and sorted by sex, physiological state and taxa. *An. gambiae* s.l and *An. funestus* s.l mosquitoes were separated in labelled tubes and analysed to identify sibling species, based on DNA from hind limbs of individual mosquitoes, processed by PCR amplification [33,34].

## Detection *Plasmodium* sporozoite infection in malaria vectors

Female *Anopheles* mosquitoes caught in Suna and RBu traps were segregated by taxa and examined in pools of ten, using ELISA assays to detect *Plasmodium* circumsporozoite antigens in the mosquito salivary glands [35].

## Identification of mosquito blood-meal hosts

The blood-fed *Anopheles*, which were mainly caught by the RBu-traps, were screened for presence of immunoglobulins indicative of human, dog, chicken, bovine and goat in the mosquito abdomen [36].

## Ethics statement

Volunteers participating in this study were adequately informed of the study objectives, potential benefits and potential risks, after which written informed consents were obtained. Since we used the odour-baited mosquito and Resting bucket traps, the volunteers were not at risks of being infected with malaria parasites during the experiments. Ethical approval was obtained from the institutional review board of Ifakara Health Institute (IHI/IRB/No: 34–2014) and the Medical Research Coordinating Council at Tanzanian National Institute of Medical Research Certificate No. NIMR/HQ/R.8a/Vol.IX1903). This manuscript has also granted a permission to publish by NIMR, reference number; NIMR/HQ/P.12 VOL XXX/.

## Data analysis

Mosquito count data was analysed using R-statistical software [37]. Factors affecting distribution of malaria vectors were investigated using generalised additive mixed-effect models (*gamm*) with a Poisson distribution, separately for each mosquito species. Only host-seeking

mosquito count data were used to assess the association of residual malaria vectors and factors affecting their distributions. However, resting mosquito data were subjected to descriptive statistics which provided their composition in proportions. Hot-seeking mosquito counts were modelled as a function of land use, environmental, anthropogenic and distance-related factors. To account for the effect of time, a smoothing function of the month (1 to 12) was included as a cyclic cubic spline. To assess consistent patterns in the spatial autocorrelation of the host-seeking mosquitoes counts, a bivariate tensor-product P-spline of X and Y coordinates of the centroid of each cell was used [38]. A random effect was included on cell label, because some of the cells were sampled more than once.

The first models included all measured variables, but Akaike's Information Criterion (AIC) was performed to select the best subset of variables for each host-seeking mosquito species. Model fitness was assessed by graphically inspecting residuals versus fitted plot to verify homogeneity [39]. Then, the Relative risk ratio (RR) together with 95% confidence intervals (CI) was calculated from the model estimates.

Using the best model for each host-seeking mosquito species, each sampled cell was reclassified according to observed mosquito counts, and maps showing the expected abundance for each species were generated. To visualize maps of such predictions according to a continuous surface, and to highlight "hotspots" of mosquito abundance, ordinary Kriging estimation was done [40], using the cell centroids and predicted mosquito counts. All of the malaria vectors count data caught by both Suna® traps and RBu-traps were combined to assess the species composition, blood feeding and sporozoite rates and the results were presented in percentages.

## Results

### Host-seeking and resting mosquitoes caught

A total of 8,992 *Anopheles* mosquitoes were caught, of which 90% (8,089) were unfed suggesting there were host-seeking (caught using Suna® traps) and 10% (903) were resting mosquitoes (caught using RBu traps). Of the host-seeking mosquitoes, 19.2% (1556) were *An. arabiensis*, 1.9% (155) *An. funestus* and 90% (7281) were other *Anopheles* spp (Table 1A). These RBu-traps caught a total of 405 female mosquitoes, whereby about 52.3% (210) were female *Culex spp.* followed by *Anopheles spp.* 28.5% (117) and *Mansonia spp* 19.3% (78). Of the resting mosquitoes, 44.5% (402) were female and 55.5% (501) male mosquitoes. Only 17 female *An. arabiensis* were caught by RBu-traps, of which 13 were blood-fed and 4 unfed.

### Blood-meal preferences and *Plasmodium* infection status of *Anopheles* mosquitoes

A total of 24 blood-fed female *An. arabiensis* mosquitoes were caught by both Suna and RBu traps, of which 58.3% (14) had fed on bovines, 25% (6) on humans and 16.7% (4) on dogs. All of the 8,992 female malaria vectors tested for *Plasmodium* infection turned out to be negative (Table 1B).

### Environmental and anthropogenic factors influencing malaria vector densities

Presence of water bodies covered with vegetation was associated with 22% higher densities of *An. arabiensis*, 51% lower densities of *An. funestus* and 59% lower densities of other *Anopheles* spp. Seasonal swamps also significantly increased *An. arabiensis* densities by 35%, but had no effect on the other vectors. Presence of natural water bodies such as rivers and springs were associated with increased the densities of the *An. arabiensis* (91% increase), but had no effect

**Table 1. a: Sibling species identification of primary malaria vectors. 1b: Blood-meal and sporozoite detection in *Anopheles* species caught.**

| Laboratory assay | Species tested | No. specimen | PCR-amplification rate | Species confirmed |
|---|---|---|---|---|
| **Species identification- PCR** | *Anopheles gambiae s.l.* | 1556 | 1291/1556 (82.9%) | 1291/1291 (100.0%) *An. arabiensis* |
| | *Anopheles funestus s.l.* | 155 | 133/155 (85.8%) | 71/133 (53.4%) *An. funestus Giles* |
| | | | | 55/133 (41.4%) *An. rivulorum* |
| | | | | 7/133 (5.3%) *An. leesoni* |

| Laboratory assay | Species tested | No. specimen | ELISA-detection rate | Host confirmed | Sporozoite confirmed |
|---|---|---|---|---|---|
| **Blood-meal ELISA** | *Anopheles arabiensis* | 28 | 24/28 (85.7%) | 14/24 (58.3%) Bovine | N/A |
| | | | | 6/24 (25.0%) Human | |
| | | | | 4/24 (16.7%) Dog | |
| **Sporozoite ELISA** | *Anopheles arabiensis* | 1556 | **N/A** | **N/A** | ***Plasmodium falciparum* negative** |
| | *Anopheles funestus* | 155 | | | |
| | *Anopheles coustani* | 955 | | | |
| | *Anopheles pharoensis* | 431 | | | |
| | *Anopheles squamosus* | 239 | | | |
| | *Anopheles wellcomei* | 49 | | | |
| | *Anopheles ziemanni* | 5607 | | | |
| | **Total malaria vectors** | **8992** | | | |

on other malaria vectors. Presence of grassland and shrubs around the sampling points significantly affected *An. arabiensis* and *An. funestus* densities, but slightly reduced densities of the other *Anopheles* spp. by 16%, (Table 2). Lastly, rice cultivation within 100m radius was associated with 19% lower densities of other *Anopheles* spp, but no effect was observed on densities of *An. arabiensis* or *An. funestus*.

**Table 2. Factors affecting outdoor malaria vector abundance in the three study villages in the Kilombero Valley, south-Eastern Tanzania.**

| Category | Variable | *Anopheles arabiensis* | | *Anopheles funestus* | | Other *Anopheles* species # | |
|---|---|---|---|---|---|---|---|
| | | RR [95% C.I] | P-value | RR [95% C.I] | P-value | RR [95% C.I] | P-value |
| **Environmental factors** | Grassland | 0.94 [0.82–1.08] | 0.382 | 1.14 [0.68–1.90] | 0.611 | 0.84 [0.76–0.94] | 0.001 |
| | Shrubs | 1.02 [0.89–1.17] | 0.771 | 0.67 [0.29–1.57] | 0.353 | N/A | N/A |
| | Natural water bodies | 1.91 [1.67–2.18] | <0.001 | N/A | N/A | 1.05 [0.96–1.15] | 0.023 |
| | Artificial water bodies | 1.00 [0.90–1.11] | 0.944 | 1.75 [1.20–2.56] | 0.043 | N/A | N/A |
| | Covered water bodies | 1.22 [1.13–1.34] | <0.001 | 0.49 [0.33–0.73] | 0.021 | 0.41 [0.39–0.44] | <0.001 |
| | Sunlight water bodies | 0.93 [0.85–1.00] | 0.007 | 0.52 [0.37–0.76] | 0.025 | 0.88 [0.80–1.00] | <0.001 |
| | Seasonal swamp | 1.35 [1.21–1.51] | <0.001 | N/A | N/A | 1.01 [0.94–1.08] | 0.771 |
| | Turbid water bodies | N/A | N/A | 1.68 [1.12–2.54] | 0.032 | 1.26 [1.19–1.33] | <0.001 |
| | Dirty water bodies | 0.57 [0.52–0.63] | <0.001 | N/A | N/A | 1.15 [1.08–1.21] | <0.001 |
| | Open water wells | 1.39 [1.28–1.52] | <0.001 | N/A | N/A | 1.23 [1.16–1.30] | <0.001 |
| | Wetland | 1.24 [1.14–1.34] | <0.001 | 0.73 [0.52–1.02] | 0.062 | 1.59 [1.51–1.68] | <0.001 |
| **Land use** | Agriculture (rice-field) | 0.94 [0.86–1.01] | 0.253 | 1.24 [0.85–1.79] | 0.252 | 0.91 [0.87–0.96] | <0.001 |
| | Number of chickens | 1.08 [1.04–1.12] | <0.001 | 0.84 [0.71–1.00] | 0.051 | N/A | N/A |
| | Number of houses | 0.85 [0.81–0.88] | <0.001 | N/A | N/A | 0.75 [0.72–0.77] | <0.001 |
| **Human activities** | Number of People | N/A | N/A | 0.59 [0.44–0.80] | 0.012 | 1.10 [1.06–1.14] | <0.001 |
| **Distance (m)** | Trap to nearest house | 0.84 [0.81–0.88] | <0.001 | 0.83 [0.66–1.04] | 0.122 | 1.0 [0.96–1.04] | 0.933 |

RR, Relative risks ratio at the 95% CI, #, stands for the other *Anopheles* spp. such as *An. coustani*, *An. ziemanni*, *An. pharoensis* and *An. wellcomei* and **N/A**: represent the variables which were not selected during the selection of the best models.

Sampling grids where people kept chickens had 8% more *An. arabiensis* but 16% less *An. funestus* than grid points with no chickens. Higher house densities were also associated with reduced *Anopheles* densities (Table 2). However, *An. arabiensis* catches dropped by 16% for every 1m distance between Suna® trap locations and the nearest house. There was no observable relationship between the trap-house distances and densities of either *An. funestus* or the other *Anopheles* species. Similarly, presence of people outdoors influenced the number of *Anopheles* caught. It reduced *An. funestus* densities by up to 41%, but increased densities of other *Anopheles* spp. by 10%.

### Temporal patterns of malaria vector densities

Similar temporal patterns were observed for *An. arabiensis* and other *Anopheles* species. In the first three months (February to May) there were fewer *An. arabiensis* compared to *An. funestus*. From the end of May to September, i.e. after the heavy rains and the start of dry season, there was a general increase in number of malaria vectors (Fig 3). This was followed by a steep decline of vector densities from October to December, which corresponds to the dry season, and an increase from end of December to February, when the short rains began, especially for *An. arabiensis*.

### Spatial patterns of mosquito distribution

Fig 4 shows the interpolated mosquito counts from the Kriging estimations.

There were similar patterns for both the dominant malaria vectors and other *Anopheles* spp. present in the study villages. A degree of fine-scale heterogeneity was observed between villages, and between species. In Kivukoni village, the hotspots of *Anopheles* species were small, patchy and isolated, whereas in Minepa, they were larger and uniformly distributed. Mavimba village had only intermediate spatial clustering.

In terms of predicted abundance, *An. funestus* showed the lowest expected trap's nightly mosquito count, ranging between 0–2. *An. arabiensis* count was expected to be between 1–14 mosquitoes per trap per night, whereas the highest abundance was expected for the other *Anopheles* spp., with expected values ranging from 2–167 (Fig 4). Some differences between the villages emerged in terms of expected abundance, but also for spatial distribution and clustering. In Kivukoni, in fact, besides showing relatively lower abundance for all the considered species, the hotspots seemed to be patchy and more concentrated at the external margins of the village, whereas the central part were characterized by lower abundance. In Minepa high expected counts were observed, with a homogeneous distribution of the vectors without major differences between the central and the marginal part of the village. This was particularly notable for *An. arabiensis* and *An. funestus*. In Mavimba, the expected counts were at an intermediate level compared to the other two villages, and the spatial distribution appeared to be patchier and more fragmented. Effects of different ecological variables on vector species distribution are graphically shown in the Supplementary material.

## Discussion

Fine-scale surveillance of vector-borne diseases provides important information needed for effective control. Spatial and temporal data allow researchers to draw risk maps, which can be used to predict where and when the disease or disease-vectors will be highest. The present study relied on using exiting odour-baited Suna® trap for outdoor sampling of host-seeking mosquitoes and the RBu trap for sampling outdoor resting mosquitoes which are the proven outdoor mosquito sampling tools [25,31]. The present study assessed both spatial and temporal distribution of residual malaria vectors, the malaria transmission risk, and important

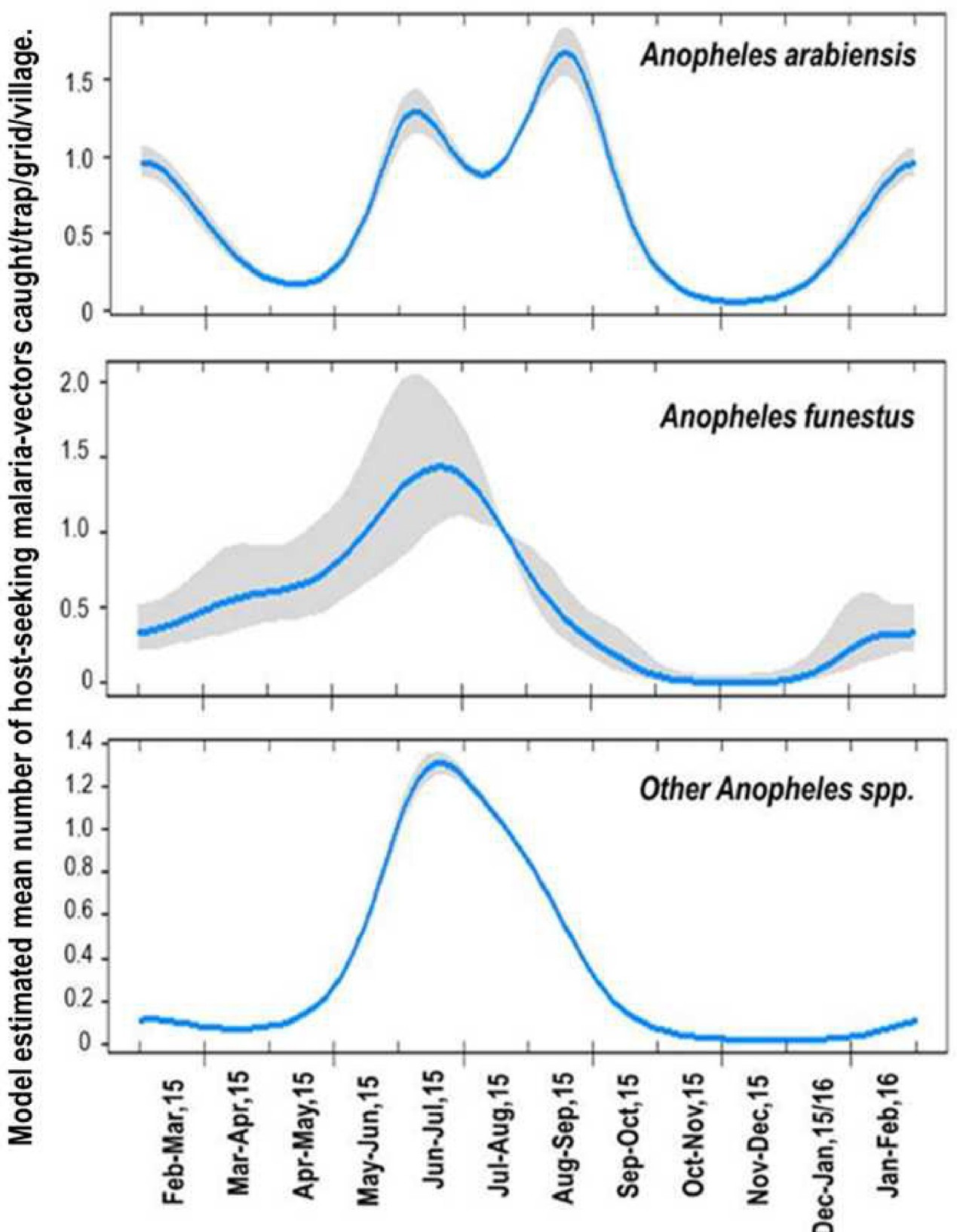

**Fig 3. Seasonal variation of abundances of outdoor host-seeking malaria vectors by month; as predicted by generalized additive mixed-effect model (GAMM) model.**

ecological factors in three low-altitude villages in rural Tanzania where LLINs are already widely used. The study examined associations between vector densities and multiple environmental, and anthropogenic factors over space and seasons.

Though the initial models included multiple factors as observed, only few were selected in the best models of mosquito abundance. Presence of water bodies covered by vegetation, such as grass and trees, showed a positive association with *An. arabiensis* vector density, which supports previous observations that this species prefers to breed in temporary shaded water bodies, such as rice paddies [41,42] (S6 Fig). These temporary water bodies were also associated with reduced *An. funestus* and other *Anopheles* spp. densities, which tend to prefer permanent water bodies with emergent vegetation [43]. Indeed, *An. funestus* in this region is now known to prefer permanent or semi-permanent water with emergent vegetation situated at least 100m from the human dwellings [44] (S4 Fig). The map grids point comprised of season swamps has positive associations with *An. arabiensis* mosquito densities as previously described by (Mala and Irungu) [42] (S7 Fig). Grids consisting of open water wells were shown to have higher numbers of *An. arabiensis* and other *Anopheles* spp, most likely because these wells provided temporary oviposition sites for the mosquitoes, as shown by previous studies [42] (S10 Fig). Similarly, wetlands did not seem to affect *An. funestus* mosquito densities, as this species prefers to breed in the permanent and deep water bodies [43,44] (S11 Fig) (*Table 2 & Supplementary materials*).

At these fine-scale resolutions, rice cultivation was not shown to affect *An. arabiensis* densities, though it reduced the abundance of the other *Anopheles* spp. Although not confirmed by statistical significance, the presence of rice paddies was associated with an increase of *An. funestus* densities, (Table 2), (S12 Fig). One possible explanation for this is that most of the rice fields found in the study area were closer to human houses, and proximity to human settlements has been observed as a factor favouring *An. funestus* breeding habitats [44]. The rice cultivation performed in this study area involves a considerable use of pesticides, which are also likely to have an effect on mosquitoes susceptibility to insecticides [45]. The *An. funestus* mosquitoes are known to exhibit high resistance level towards pyrethroids which is an active ingredient present in many pesticide [3,46], possibly confounding these results.

As shown in a similar study conducted in the same area and time [47], densities of small livestock such as chickens in the map grid significantly increased *An. arabiensis* mosquito densities indoors (S13 Fig). The correlation between the *An. arabiensis* and chicken densities in the map grid point was partly due to feeding behaviour of these mosquito species on livestock such as chickens. However, the relationship between chicken densities and malaria mosquitoes was inverse for *An. funestus* mosquitoes, thus contradicting previous studies [8,47]. It was hard to estimate the effect of the number of chickens in relation to *An. funestus* densities due to the lower population of these mosquito species outdoors as they prefers to bite and rest indoors [48].

There was a negative association between the number of people outdoors and the number of *An. funestus* mosquitoes caught, (S15 Fig). This may be partly due to competition between the host-seeking traps and humans in the vicinity, as this species has a high preference for feeding on humans over other vertebrates [48]. However, for the other *Anopheles* spp., there was an increase of the number of mosquitoes during the trapping nights with more people outdoors. This difference can be explained by the fact that the other *Anopheles* spp. are opportunistic feeders, as they feed on both human and other hosts, and are also more attracted by

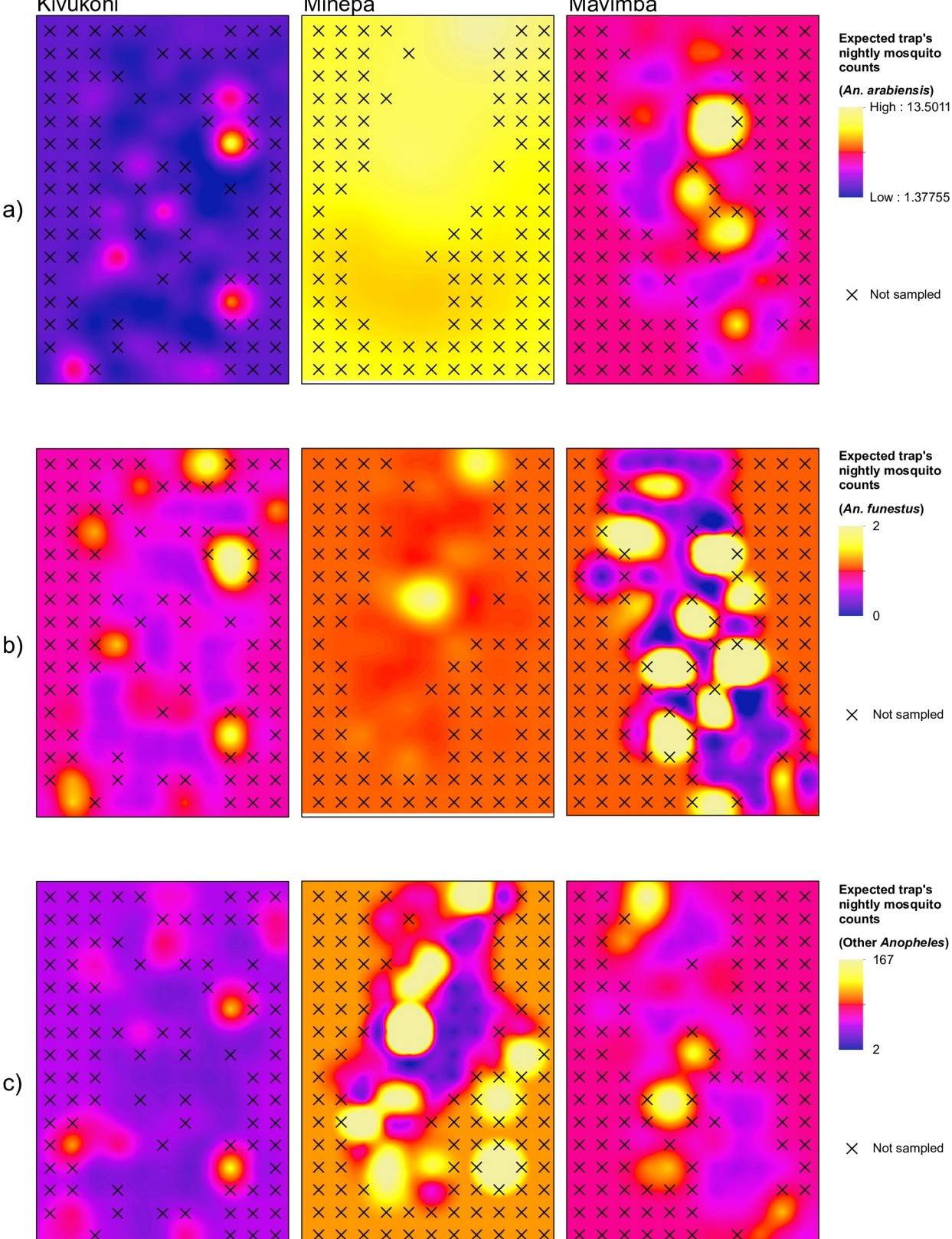

**Fig 4.** Interpolated densities of female *Anopheles* mosquitoes in the three study villages: a) *An. arabiensis*, b) *An. funestus*, c) other *Anopheles*.

synthetic attractants such as those used in the Suna® trap [49]. Also, these other *Anopheles* spp showed high potential of maintaining malaria transmission at some settings where major malaria vectors are limited [50,51]. The negative effect of distance from the houses observed in the models suggests that the main malaria vectors, *An. arabiensis* and *An. funestus* are generally concentrated near dwellings houses, where humans and cattle also reside (Table 2, S16 Fig).

The abundance of both *An. arabiensis* and the other *Anopheles* species was highest just after the long rains and the beginning of the dry season (June-September). A possible explanation might be that heavy rains, in addition to creating breeding habitats for the mosquitoes, also washed away the aquatic stages of the mosquitoes (Fig 3). This was also seen in the previous study conducted in Italy that heavy rainfall had negative effect with host-seeking behaviour of *Aedes albopictus* [52]. *An. funestus* however exhibited less fluctuation compared to other malaria vectors, most likely due to its preference for semi-permanent or permanent habitats [44]. Also, this is another sign of heterogeneity of malaria vector densities at small scale and that there may be more factors influencing these densities than we assessed.

A small proportion of female resting mosquitoes were caught by using RBu-traps, majority were Culicines and a small proportion of Anopheline mosquitoes. Blood-fed *An. arabiensis* mosquitoes were mostly fed on bovine hosts followed by human and dog hosts. In this study area, *Culex pipiens* complex was previously found in large proportion and contributed 79% of the overall indoor biting [53]. The *Culex pipiens* complex has ability to exhibit a wide range of host blood-meals [54]. All of the malaria vectors subjected to *P. falciparum* ELISA detection were found to be negative. This finding is different from previous studies which involve sampling indoors in the same villages and reported the presence of parasite positive *An. arabiensis* and *An. funestus* mosquitoes [55]. The high proportion of non-human blood-meals in the *Anopheles* mosquitoes as well as the lack of *Plasmodium*-infected specimen even after 12 months of sampling, suggests that the risk of malaria transmission in the outdoor environments in these specific villages is very low. If interpreted alongside previous studies, which found infected mosquitoes indoors [55], it can be concluded that the majority of the residual transmission events in the villages actually happens inside houses. Also this might be caused by our outdoor sampling methods which mainly caught younger mosquitoes which are most likely to be uninfected compared to the older mosquitoes [56]. It is also known that young mosquitoes are found in close proximity to the breeding habitats, hence collection of young mosquitoes might have been influenced by the presence of breeding sites in our study areas [57]. It is important therefore to continue improving indoor vector control tools, e.g. LLINs, IRS and mosquito-proof housing as the primary interventions against malaria. In addition, larval source management may be effective for controlling mosquitoes in areas where the habitats can be clearly identified.

Unlike other studies relying on remotely sensed images of climatic and environmental data, this study has demonstrated fine-scale spatial and temporal patterns of *Anopheles* mosquitoes in the outdoor environments, relying on empirical mosquito trap data and directly observed environmental and human factors. The resulting trends can aid improvements in the prediction of transmission risks even in low-malaria endemic communities where the transmission is heterogeneous.

## Conclusion

Outdoor densities of both host-seeking and resting *Anopheles* mosquitoes had significant heterogeneities between and within villages, and were influenced by multiple environmental and anthropogenic factors. Despite the high *Anopheles* densities outside dwellings, the high

proportion of non-human blood-meals and absence of malaria-infected mosquitoes after 12 months of nightly trapping suggests very low-levels of outdoor malaria transmission in the villages. It is important therefore to continue improving indoor vector control tools, e.g. LLINs, IRS and mosquito-proof housing as the primary interventions against malaria. In addition, larval source management may be effective for controlling mosquitoes in areas where the habitats can be clearly identified.

## Supporting information

**S1 Fig. Map of predicted hotspots of female mosquitoes' density and presence of grassland in the sampled grid cells in three villages in Kilombero Valley, South-Eastern Tanzania.** a) *Anopheles arabiensis*, b) *Anopheles funestus*, c) other *Anopheles*.
(DOCX)

**S2 Fig. Map of predicted hotspots of female mosquitoes' density and presence of shrubs in the sampled grid cells in three villages in Kilombero Valley, South-Eastern Tanzania.** a) *Anopheles arabiensis*, b) *Anopheles funestus*, c) other *Anopheles*.
(DOCX)

**S3 Fig. Map of predicted hotspots of female mosquitoes' density and presence of natural waterbodies in the sampled grid cells in three villages in Kilombero Valley, South-Eastern Tanzania.** a) *Anopheles arabiensis*, b) *Anopheles funestus*, c) other *Anopheles*.
(DOCX)

**S4 Fig. Map of predicted hotspots of female mosquitoes' density and presence of artificial waterbodies in the sampled grid cells in three villages in Kilombero Valley, South-Eastern Tanzania.** a) *Anopheles arabiensis*, b) *Anopheles funestus*, c) other *Anopheles*.
(DOCX)

**S5 Fig. Map of predicted hotspots of female mosquitoes' density and presence of covered waterbodies in the sampled grid cells in three villages in Kilombero Valley, South-Eastern Tanzania.** a) *Anopheles arabiensis*, b) *Anopheles funestus*, c) other *Anopheles*.
(DOCX)

**S6 Fig. Map of predicted hotspots of female mosquitoes' density and presence of sunlight waterbodies in the sampled grid cells in three villages in Kilombero Valley, South-Eastern Tanzania.** a) *Anopheles arabiensis*, b) *Anopheles funestus*, c) other *Anopheles*.
(DOCX)

**S7 Fig. Map of predicted hotspots of female mosquitoes' density and presence of seasonal swamps in the sampled grid cells in three villages in Kilombero Valley, South-Eastern Tanzania.** a) *Anopheles arabiensis*, b) *Anopheles funestus*, c) other *Anopheles*.
(DOCX)

**S8 Fig. Map of predicted hotspots of female mosquitoes' density and presence of turbid waterbodies in the sampled grid cells in three villages in Kilombero Valley, South-Eastern Tanzania.** a) *Anopheles arabiensis*, b) *Anopheles funestus*, c) other *Anopheles*.
(DOCX)

**S9 Fig. Map of predicted hotspots of female mosquitoes' density and presence of dirty waterbodies in the sampled grid cells in three villages in Kilombero Valley, South-Eastern Tanzania.** a) *Anopheles arabiensis*, b) *Anopheles funestus*, c) other *Anopheles*.
(DOCX)

**S10 Fig. Map of predicted hotspots of female mosquitoes' density and presence of open water well in the sampled grid cells in three villages in Kilombero Valley, South-Eastern Tanzania.** a) *Anopheles arabiensis*, b) *Anopheles funestus*, c) other *Anopheles*.
(DOCX)

**S11 Fig. Map of predicted hotspots of female mosquitoes' density and presence of wetland in the sampled grid cells in three villages in Kilombero Valley, South-Eastern Tanzania.** a) *Anopheles arabiensis*, b) *Anopheles funestus*, c) other *Anopheles*.
(DOCX)

**S12 Fig. Map of predicted hotspots of female mosquitoes' density and presence of rice fields in the sampled grid cells in three villages in Kilombero Valley, South-Eastern Tanzania.** a) *Anopheles arabiensis*, b) *Anopheles funestus*, c) other *Anopheles*.
(DOCX)

**S13 Fig. Map of predicted hotspots of female mosquitoes' density and number of chickens in the sampled grid cells in three villages in Kilombero Valley, South-Eastern Tanzania.** a) *Anopheles arabiensis*, b) *Anopheles funestus*, c) other *Anopheles*.
(DOCX)

**S14 Fig. Map of predicted hotspots of female mosquitoes' density and number of houses in the sampled grid cells in three villages in Kilombero Valley, South-Eastern Tanzania.** a) *Anopheles arabiensis*, b) *Anopheles funestus*, c) other *Anopheles*.
(DOCX)

**S15 Fig. Map of predicted hotspots of female mosquitoes' density and number of people in the sampled grid cells in three villages in Kilombero Valley, South-Eastern Tanzania.** a) *Anopheles arabiensis*, b) *Anopheles funestus*, c) other *Anopheles*.
(DOCX)

**S16 Fig. Map of predicted hotspots of female mosquitoes' density and distance from trap to the nearest house in the sampled grid cells in three villages in Kilombero Valley, South-Eastern Tanzania.** a) *Anopheles arabiensis*, b) *Anopheles funestus*, c) other *Anopheles*.
(DOCX)

## Acknowledgments

A special thanks to volunteers who allowed us to install the traps at their premises and the ones who involved in trapping and retrieving the mosquito traps during the study. Many thanks to Miss Claudia Eichenberger for her critical review of this manuscript prior submission, and to Miss Keila Meginnis for her English proofreading and editing suggestions. Mr. Godfrey P. Lingamba died before the submission of the final version of this manuscript. Arnold Sadikiel Mmbando accepts responsibility for the integrity and validity of the data collected and analysed.

## Author Contributions

**Conceptualization:** Fredros O. Okumu.

**Data curation:** Arnold S. Mmbando, Fredros O. Okumu.

**Formal analysis:** Arnold S. Mmbando, Halfan S. Ngowo, Luca Nelli.

**Funding acquisition:** Fredros O. Okumu.

**Investigation:** Arnold S. Mmbando, Fredros O. Okumu.

**Methodology:** Arnold S. Mmbando, Johnson K. Swai, Nancy S. Matowo, Masoud Kilalangongono, Godfrey P. Lingamba, Joseph P. Mgando, Fredros O. Okumu.

**Resources:** Fredros O. Okumu.

**Supervision:** Arnold S. Mmbando, Emmanuel W. Kaindoa, Masoud Kilalangongono, Godfrey P. Lingamba, Joseph P. Mgando, Fredros O. Okumu.

**Visualization:** Fredros O. Okumu.

**Writing – original draft:** Arnold S. Mmbando, Johnson K. Swai, Nancy S. Matowo, Isaac H. Namango, Fredros O. Okumu, Luca Nelli.

**Writing – review & editing:** Arnold S. Mmbando, Emmanuel W. Kaindoa, Halfan S. Ngowo, Johnson K. Swai, Nancy S. Matowo, Isaac H. Namango, Fredros O. Okumu, Luca Nelli.

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
