## [Decision Letter · Decision Letter 0]

5 Aug 2020

PONE-D-20-16651

Fine-scale distribution of malaria mosquitoes biting or resting outside human dwellings in three low-altitude Tanzanian villages

PLOS ONE

Dear Dr. Mmbando,

Thank you for submitting your manuscript for review to PLoS ONE.After careful consideration, we have concluded that your manuscript requires substantial revision, following which it can possibly be reconsidered. According to reviewers   some statements   were not supported by the manuscript   findings. Study design should be clarified and conclusion must be restricted to the authors’ findings. As quoted by the reviewer #1, for example,  only 28 blood-fed mosquitoes may not allow concluding about the feeding habits of a vector. It is possible that inappropriate collection sites may have influenced the results.   Reviewer #2 complains that the picture showed in the paper is more linked to host-seeking and not resting mosquitoes because of the high number of the host seeking versus resting mosquitoes. For your guidance, a copy of the reviewers' comments was included below

We look forward to receiving your revised manuscript.

Kind regards,

Luzia Helena Carvalho, Ph.D.

Academic Editor

PLOS ONE

Journal Requirements:

3. We note that [Figure(s) 1] in your submission contain [map/satellite] images which may be copyrighted. All PLOS content is published under the Creative Commons Attribution License (CC BY 4.0), which means that the manuscript, images, and Supporting Information files will be freely available online, and any third party is permitted to access, download, copy, distribute, and use these materials in any way, even commercially, with proper attribution. For these reasons, we cannot publish previously copyrighted maps or satellite images created using proprietary data, such as Google software (Google Maps, Street View, and Earth). For more information, see our copyright guidelines: http://journals.plos.org/plosone/s/licenses-and-copyright.

1.    You may seek permission from the original copyright holder of Figure(s) [1] to publish the content specifically under the CC BY 4.0 license. 

Reviewers' comments:

Reviewer's Responses to Questions

**Comments to the Author**

1. Is the manuscript technically sound, and do the data support the conclusions?

Reviewer #1: Partly

Reviewer #2: Yes

2. Has the statistical analysis been performed appropriately and rigorously? 

Reviewer #1: I Don't Know

Reviewer #2: Yes

3. Have the authors made all data underlying the findings in their manuscript fully available?

Reviewer #1: Yes

Reviewer #2: Yes

4. Is the manuscript presented in an intelligible fashion and written in standard English?

Reviewer #1: Yes

Reviewer #2: Yes

5. Review Comments to the Author

Reviewer #1: Fine-scale distribution of malaria mosquitoes biting or resting outside human dwellings in three low-altitude Tanzanian villages

It is generally acknowledged that a greater understanding of the distribution and dynamics of malaria vectors outside of houses is becoming increasingly important and so efforts such as this paper are useful. The methods used in the present paper were sufficiently rigorous for possible conclusions to be determined. I am not sure, however, that despite their best efforts the authors have been able to contribute anything of significance to the scientific canon.

Overall a total of 28 blood-fed mosquitoes is hardly sufficient to draw conclusions about the feeding habits of a mosquito (even though they fit within the accepted framework of what An. arabiensis will feed on) – to quote percentages in the abstract (without providing sample size) is specious.

One problem might be that the collection sites were determined from a map rather than from the ground. Thus, potentially high-density areas may not have been sampled even though they might have been identified by the researchers during a preliminary visit or two. For example, if I am correct the village of Kivukoni is close to the Kilombero River. Previous work has indicated that densities of insects (particularly An. arabiensis) are very high close to the river margins (where the old ferry used to cross) but I notice that this was not actually sampled in the present study.

My own opinion (and it is no more than an opinion) is that in the future, following interventions that significantly reduce vector densities, sampling should be undertaken in high density locations (‘hot spots’ as described by the authors). I am not sure that studies like theirs enable the easy identification of such hot spots, which presumably exist everywhere.

On another point it has been suggested that younger insects are more likely to bite outdoors compared to older ones (e.g. PeerJ 5155) which may explain the lack of infected insects (but which might also mean that targeting such insects would be a useful control technique).

Given that rainfall patterns in many parts of the world are now less predictable than they used to be an indication of the actual rainfall observed during the study (in one of the villages at least) would be useful rather than merely saying that densities were highest at the end of the wet season. Both of the vectors are known to show such patterns – and indeed they have been described (with included rainfall) from areas close to the study villages.

As I understand it the efficacy of the Suna trap is dependent on the placement of the trap in relation to other objects and/or hosts. Thus, whilst putting the traps in the centroids of their sampling grid makes sense from one perspective it might mean that the efficiency (and therefore estimated estimate of population density) is affected. A more concentrated study (with traps in a finer scale grid) might help evaluate this.

The other thing that is, of course, missing from the study is an evaluation of the indoor density of mosquitoes in the study villages. Perhaps the authors undertook such sampling but are hoping to publish this data at another time. As far as I know Suna traps have not yet been used indoors but there is no reason why they should not be used in this way. The elegant stand shown could be placed inside a house to obtain an equivalent sample to the outdoor one.

Although a number of other species have been identified as being possible vectors (such as their reference #5) it has not been suggested that these ‘can become important’ in the absence of the primary vectors. What they may do is maintain a low level of transmission that might result in an epidemic if the principal vectors return (following say the decline in insecticidal effect of an IRS treatment).

The authors state that ‘…. presence of people outdoors influenced the number of Anopheles caught. It reduced An. funestus densities by up to 41%, but increased densities of other Anopheles spp. by 10%’. I would suggest that the reduction in the numbers of the anthropogenic An. funestus in their traps was because the insects were off biting people under those circumstances and were at the same time increasing the number of catholic feeders such as the non-vectors which (by being attracted to the carbon dioxide more than anything else) would then also be caught in the trap.

There are a number of relatively trivial corrections required to the English in the manuscript.

Reviewer #2: GENERAL COMMENTS

The micro geographic level, the transmission is no the same from house to another house. The factor explaining this micro variation is not well understood. Therefore, the problematic is interesting and deserves to be raised. Understanding this variation facilitates delivery of targeted, cost-effective preventative antivectorial interventions against malaria. The paper is well written and the methodology is well-designed to address the question. However, some part of the paper as presented, needs revisions to make it more precise and challenging:

ABSTRACT: Fine-scale distribution of malaria mosquitoes biting or resting outside human dwellings in three low-altitude Tanzanian villages

BACKGROUND

Line 2: long-lasting insecticidal bed nets (LLINs) : add the IRS in the list of prevention measures that contribute to reduce malaria cases during the last decade

“…….notably mosquito resistance to commonly used insecticides…” insert a reference confirming the statement.

MATERIAL AND METHODS

Selection of outdoor mosquito sampling units

“Mosquitoes were sampled from each of these sentinel grid points for ten nights each round (totalling 30 trap-nights/month)….” Check this sentence it is no clear. you mentioned 30 trap-nights/month. You have two sentinel sites per village. One round is ten days. I suppose there is one trap per sentinel sites. I guess the total trap per site is 20trap-nights so for the 3 sites you are 60 trap-nights the month

Data analysis

Even it is not state in the paper, the analysis is done by pooling outdoor host-seeking mosquito's fraction and those resting outside. I don’t understand this rational. I suggest to split the two populations and look how the different parameters influence host seeking and resting habits separately. The way the author pool the two population makes some confusion because the host-seeking mosquitoes can bite and goes to rest indoor. And also the outdoor resting mosquitoes may come from the indoor biting fraction. I understand that the number of malaria vector specimens are very low in the resting population but we can’t pool. The picture you showed in the paper is more linked to host-seeking and not resting mosquitoes because of the high number of the host-seeking (all mosquitoes: 8089 host-seeking versus 903 resting and vectors (arabiensis and funestus):1556+155 host seeking versus 17 resting))

FIGURES AND LEGEND

Figure 1: Study area: I suggest to show in the map the location of the two sentinel sites within each village

Figure 3: the legend of the Y axis need more precision (Mean number of malaria vector per trap caught per month???)

Figure 4: is unclear, because the resolution is low. Please improve it.

6. PLOS authors have the option to publish the peer review history of their article (what does this mean?). If published, this will include your full peer review and any attached files.

Reviewer #1: **Yes: **J D Charlwood

Reviewer #2: No

---

## [Author Response · Author response to Decision Letter 0]

23 Sep 2020

12th August 2020

Dear Editor, PLOS ONE

Re: Research paper re-submission (Manuscript code: PONE-D-20-16651)

We thank the reviewers for reviewing this manuscript titled “Fine-scale distribution of malaria mosquitoes biting or resting outside human dwellings in three low-altitude Tanzanian villages”. We kindly thank reviewers and Editor for their constructive comments on this manuscript. Please find all the comments/suggestion addressed in the revised version of the manuscript. All of the edits were highlighted in yellow which can now be included in the new version of the manuscript ready for publication. 

In short all of the edits were made as suggested by the reviewers as point by point below, which include rephrasing the sentences, figures improvements, data analysis section, adding references, clarifying some details in the experimental setups and improving the discussion of the findings and compare it with previous studies and all these changes were left highlighted in yellow in the revised manuscript. 

Once again thank you so much for your support and hope to hear from you. 

Yours sincerely

Arnold 

ammbando@ihi.or.tz

Response to reviewer’s comments

Reviewer #1:

It is generally acknowledged that a greater understanding of the distribution and dynamics of malaria vectors outside of houses is becoming increasingly important and so efforts such as this paper are useful. The methods used in the present paper were sufficiently rigorous for possible conclusions to be determined. I am not sure, however, that despite their best efforts the authors have been able to contribute anything of significance to the scientific canon.

Response: Thank you so much for the above concern, and we totally agree with you on the need to accelerate effort on outdoor mosquito control which at some settings mediate even lower level of malaria and other mosquito-borne illnesses. Also, by considering the limited resource countries like Tanzania, malaria control campaigns need to rely on small/little resource we have to fight against malaria which still kills a lot of <5 years old and pregnant women in our country. To do this we need to know exactly location and time to target these malaria-vectors at fine scale and even to predict on mosquito densities based on time and space. Thus, this study brings an insight to the focal malaria vector control experts so that they know where and when to apply vector-control interventions with limited resources we have rather than focusing on the entire villages which will consume a lot of resources and time. 

Overall a total of 28 blood-fed mosquitoes is hardly sufficient to draw conclusions about the feeding habits of a mosquito (even though they fit within the accepted framework of what An. arabiensis will feed on) – to quote percentages in the abstract (without providing sample size) is specious.

Response: Thank you so much for bring this up. We have now added the sample size in the abstract section, and acknowledged the limitations of having these small numbers. Please find it in the result section, page 1, lines 62-63. 

One problem might be that the collection sites were determined from a map rather than from the ground. Thus, potentially high-density areas may not have been sampled even though they might have been identified by the researchers during a preliminary visit or two. For example, if I am correct the village of Kivukoni is close to the Kilombero River. Previous work has indicated that densities of insects (particularly An. arabiensis) are very high close to the river margins (where the old ferry used to cross) but I notice that this was not actually sampled in the present study.

Response: Thank you this comment. The target of this study was to assess the spatial and temporal distribution patterns of malaria vectors outdoors for which the sampling location was selected based on the previous study conducted in the same villages by Mwangungulu et al 2013. To be able to cover the entire study villages map grip points were first selected based on the settlement patterns which are concentrated at the middle of the village which also where most of mosquitoes are concentrated. The grid points which met the inclusion criterial for mosquito sampling were then randomly selected in such a way that they represent the entire study villages to avoid the positional biasness. Since, along the river margins where no/fewer human dwellings led us to have fewer sampling locations as compared to the center of Kivukoni village. This is well explained in the method section in page 5 first paragraph, lines 142-148. 

My own opinion (and it is no more than an opinion) is that in the future, following interventions that significantly reduce vector densities, sampling should be undertaken in high density locations (‘hot spots’ as described by the authors). I am not sure that studies like theirs enable the easy identification of such hot spots, which presumably exist everywhere.

Response: Thank you for the above explanation. This study was designed to also capture the variations of vector densities across the sampled grids/areas at a specific time. We agree that sampling should be done in high mosquito density areas, but first we need to identify those high density areas, which was the exactly the target of this study. This is clearly found in the method section page 5, lines 142-157. 

On another point it has been suggested that younger insects are more likely to bite outdoors compared to older ones (e.g. PeerJ 5155) which may explain the lack of infected insects (but which might also mean that targeting such insects would be a useful control technique).

Response: Thank you very much for this comment. This is a valid point. We have added a paragraph under the discussion section to capture the concept that younger insects are more likely to bite outdoors compared to older ones hence our study might have been affected by the outdoor collection which may have led to limited collection of infected mosquitoes. This also was well explained in a separate study done in the same study area at the same time which showed more malaria transmission occurred indoors and it was mostly mediated by Anopheles funestus mosquitoes and small extent Anopheles arabiensis, (Kaindoa et al 2018). The combination of these two surveillances could answer why most of the transmission occurred indoors compared to the outdoor. See addition information in the discussion section page 15, lines 411-415. 

Given that rainfall patterns in many parts of the world are now less predictable than they used to be an indication of the actual rainfall observed during the study (in one of the villages at least) would be useful rather than merely saying that densities were highest at the end of the wet season. Both of the vectors are known to show such patterns – and indeed they have been described (with included rainfall) from areas close to the study villages.

Response: Thank you so much for pointing this out. We have now added the rainfall data gathered from the previous study (Ngowo et al 2017) done at the same time at 30km from Kivukoni village. See it in page 4, lines 131-133. 

As I understand it the efficacy of the Suna trap is dependent on the placement of the trap in relation to other objects and/or hosts. Thus, whilst putting the traps in the centroids of their sampling grid makes sense from one perspective it might mean that the efficiency (and therefore estimated estimate of population density) is affected. A more concentrated study (with traps in a finer scale grid) might help evaluate this.

Response: Thank you for this. We did dot evaluate the impact of Suna trap on mosquito collection based on the finer scale as we relied on the previous evidence on the efficacy of Suna trap for outdoor mosquito sampling, however, we have other separate studies which are currently ongoing which evaluate the effectiveness of different outdoor mosquito sampling tools which are conducted in one village. This is well explained in page 13, first paragraph, 333-335. 

The other thing that is, of course, missing from the study is an evaluation of the indoor density of mosquitoes in the study villages. Perhaps the authors undertook such sampling but are hoping to publish this data at another time. As far as I know Suna traps have not yet been used indoors but there is no reason why they should not be used in this way. The elegant stand shown could be placed inside a house to obtain an equivalent sample to the outdoor one.

Response: Thank you reviewer for this. Suna trap use attractants hence it is discouraged to put mosquito attractants inside human houses. Ethically it is not acceptable to attract mosquitoes toward humans inside the houses. Furthermore, adding attraction inside human dwellings may overestimate the indoor vector densities. However, we do have another study by which was done to assess indoor vector densities using CDC LT and Prockpack. (Kaindoa et al 2018) which was done in the same study village at the same time which assessed the impact of indoor residual malaria vectors as well as how settlements and human biomass patterns affect mosquito densities, (Kaindoa et al 2018). See it in page 14, lines 369-371. 

Although a number of other species have been identified as being possible vectors (such as their reference #5) it has not been suggested that these ‘can become important’ in the absence of the primary vectors. What they may do is maintain a low level of transmission that might result in an epidemic if the principal vectors return (following say the decline in insecticidal effect of an IRS treatment).

Response: We have added a paragraph in the discussion section to highlight the potential of other vector species in maintaining malaria transmission though at lower levels. See it page 14, lines 384-386. 

The authors state that ‘…. presence of people outdoors influenced the number of Anopheles caught. It reduced An. funestus densities by up to 41%, but increased densities of other Anopheles spp. by 10%’. I would suggest that the reduction in the numbers of the anthropogenic An. funestus in their traps was because the insects were off biting people under those circumstances and were at the same time increasing the number of catholic feeders such as the non-vectors which (by being attracted to the carbon dioxide more than anything else) would then also be caught in the trap.

Response: Thank you for the above concern. We agree with you that, there were competition in terms of mosquito attractions between carbon dioxide baited Suna-traps and humans outdoors during the sampling time. This is more seen in An. funestus mosquitoes than other malaria vectors because these mosquitoes are high anthropophilic as well as they prefer to bite indoors than outdoors. Other, malaria mosquitoes such as An. arabiensis and secondary vectors are opportunistic feeders for which they had a wide range of feeding hosts. This is well explained in page 14, lines 376-384.

There are a number of relatively trivial corrections required to the English in the manuscript.

 Response: Thank for pointing out this concern. We have now revised the manuscript accordingly. 

Reviewer #2: 

GENERAL COMMENTS

The micro geographic level, the transmission is no the same from house to another house. The factor explaining this micro variation is not well understood. Therefore, the problematic is interesting and deserves to be raised. Understanding this variation facilitates delivery of targeted, cost-effective preventative antivectorial interventions against malaria. The paper is well written and the methodology is well-designed to address the question. However, some part of the paper as presented, needs revisions to make it more precise and challenging:

Response: Thank you so much for the acknowledgements and we have now revised the manuscript in point by point below

ABSTRACT: Fine-scale distribution of malaria mosquitoes biting or resting outside human dwellings in three low-altitude Tanzanian villages

BACKGROUND

Line 2: long-lasting insecticidal bed nets (LLINs) : add the IRS in the list of prevention measures that contribute to reduce malaria cases during the last decade

Response: Thank you for pointing out this. We have added IRS in the revised manuscript as suggested. Changes are found in the first paragraph, page 3, lines 77-79.

“…….notably mosquito resistance to commonly used insecticides…” insert a reference confirming the statement.

Response: Thank you reviewer for this. The reference has been added in the revised manuscript in paragraph 1 page 2, line 83. 

MATERIAL AND METHODS

Selection of outdoor mosquito sampling units

“Mosquitoes were sampled from each of these sentinel grid points for ten nights each round (totalling 30 trap-nights/month)….” Check this sentence it is no clear. you mentioned 30 trap-nights/month. You have two sentinel sites per village. One round is ten days. I suppose there is one trap per sentinel sites. I guess the total trap per site is 20trap-nights so for the 3 sites you are 60 trap-nights the month

Response: Thank you for pointing this out. It is true that we have two fixed grids point/village and four randomly selected map grid points which changed each night. We had 30 trap-nights/month because the four randomly selected grids/site/night were sampled together with the two sentinel grids points/site which totaling 6-grids points/site/night. This is well explained in page 5 lines 142-148. 

Data analysis

Even it is not state in the paper, the analysis is done by pooling outdoor host-seeking mosquito's fraction and those resting outside. I don’t understand this rational. I suggest to split the two populations and look how the different parameters influence host seeking and resting habits separately. The way the author pool the two population makes some confusion because the host-seeking mosquitoes can bite and goes to rest indoor. And also the outdoor resting mosquitoes may come from the indoor biting fraction. I understand that the number of malaria vector specimens are very low in the resting population but we can’t pool. The picture you showed in the paper is more linked to host-seeking and not resting mosquitoes because of the high number of the host-seeking (all mosquitoes: 8089 host-seeking versus 903 resting and vectors (arabiensis and funestus):1556+155 host seeking versus 17 resting))

Response: Thank you so much for pointing this up. This was added into the main manuscript that, resting malaria vectors were dropped during the analysis of anthropogenic, environmental and distance related factors impact on malaria vectors. The resting malaria vectors caught were lower in number for the model to select. We did descriptive statistics (proportions and percentage) to determine the composition of resting malaria vectors. Reason for splitting host-seeking and resting malaria vectors was due to huge difference in numbers of mosquitoes caught by these two methods. However, all of the malaria vectors caught by both Suna and Resting bucket traps were packed and submitted to the laboratory for different assays, i.e. specie ID, sporozoite analysis and blood meal ELISA. Please revised data analysis section in page 7,lines 210-234. 

FIGURES AND LEGEND

Figure 1: Study area: I suggest to show in the map the location of the two sentinel sites within each village

Response: Thank you so much for this. Please see the revised study area map, Figure 1. 

Figure 3: the legend of the Y axis need more precision (Mean number of malaria vector per trap caught per month???)

Response: Thank you for this. Please find revised Y-axis legend in the Figure 3. 

Figure 4: is unclear, because the resolution is low. Please improve it.

Response: Thank you for this. Please find high resolution of the Figure 4 in the revised manuscript. 

6. PLOS authors have the option to publish the peer review history of their article (what does this mean?). If published, this will include your full peer review and any attached files.

Do you want your identity to be public for this peer review? For information about this choice, including consent withdrawal, please see our Privacy Policy.

Reviewer #1: Yes: J D Charlwood

Reviewer #2: No

---

## [Decision Letter · Decision Letter 1]

16 Nov 2020

PONE-D-20-16651R1

Fine-scale distribution of malaria mosquitoes biting or resting outside human dwellings in three low-altitude Tanzanian villages

PLOS ONE

Dear Dr. Mmbando,

Thank you for submitting your manuscript to PLoS ONE. After careful consideration, we felt that your manuscript requires revision, following which it can possibly be reconsidered. At this time, two major concerns need to be clarified by the authors.  First, about the choice of traps that might not be the best to study outdoor transmission (bias towards the anthropophilic An. Funestus). Second, The paper would gain in clarity and robustness if the other Anopheles species were also taken into consideration, especially as the data are already available.  Finally, the authors should still address a number of minor corrections and comments raised by the reviewer #3. 

We look forward to receiving your revised manuscript.

Kind regards,

Luzia Helena Carvalho, Ph.D.

Academic Editor

PLOS ONE

Reviewers' comments:

Reviewer's Responses to Questions

**Comments to the Author**

1. If the authors have adequately addressed your comments raised in a previous round of review and you feel that this manuscript is now acceptable for publication, you may indicate that here to bypass the “Comments to the Author” section, enter your conflict of interest statement in the “Confidential to Editor” section, and submit your "Accept" recommendation.

Reviewer #2: All comments have been addressed

Reviewer #3: (No Response)

2. Is the manuscript technically sound, and do the data support the conclusions?

Reviewer #2: Yes

Reviewer #3: Yes

3. Has the statistical analysis been performed appropriately and rigorously? 

Reviewer #2: Yes

Reviewer #3: Yes

4. Have the authors made all data underlying the findings in their manuscript fully available?

Reviewer #2: Yes

Reviewer #3: Yes

5. Is the manuscript presented in an intelligible fashion and written in standard English?

Reviewer #2: Yes

Reviewer #3: Yes

6. Review Comments to the Author

Reviewer #2: One of my big concern is the pooling of hostseeking and resting population for the analysis. Now the authors have considered all comments.

Reviewer #3: This study is a sound work on the spatio-temporal analysis of the malaria vectors collected outdoors in 3 villages of Tanzania. The manuscript has already been improved after a first review process, but few concerns remain that need to be taken into consideration, especially 2 main comments.

1. The choice of traps might not be the best to study outdoor transmission, as raised by the authors themselves on lines 387-389, because host-seeking traps and humans in the vicinity introduced a bias, An. funestus being highly anthropophilic, this species will avoid the traps to feed on humans. This is particularly obvious in Table 1a in which only 155 specimens of An. funestus s.l. where collected compared to 1556 specimens of An. arabiensis. Is this small number due to the traps that were not appropriate for this species or to its low density? Therefore, the choice of the 2 types of traps should be better explained.

2. Seven mosquito taxa have been collected during the 12 months collection (Table 1b). However, the manuscript is focusing on 2 species only, An. arabiensis and An. funestus. The third category is named "Other Anopheles". The paper would gain in clarity and robustness if the other species were also taken into consideration, especially as the data are already available (Table 1b). When possible, the spatio-temporal analysis should be more specific, including for instance An. zeimanni which was collected in high numbers (n=5607 specimens), 5 times more than An. arabiensis.

They are also a good number of minor corrections and comments that have been included into the attached manuscript for its improvement (see attached file). For instance, they are some statements that need further explanation to increase their understanding.

Besides, the choice of the analysis as shown in Figure 4 and many supplementary ones (S1.1-16) is quite pertinent and well representative of the factors influencing the presence of malaria vectors.

As a suggestion in the discussion, a GIS mapping of the spatio-temporal factors influencing the presence and abundance of each malaria vector could be developed if a follow up of this work is to be done.

7. PLOS authors have the option to publish the peer review history of their article (what does this mean?). If published, this will include your full peer review and any attached files.

Reviewer #2: No

Reviewer #3: **Yes: **Sylvie MANGUIN

---

## [Author Response · Author response to Decision Letter 1]

27 Nov 2020

24th November 2020

Dear Editor, PLOS ONE

Re: Research paper re-submission (Manuscript code: PONE-D-20-16651)

We would like to thank you and the reviewers for constructive comments and for offering us the opportunity to resubmit a revised version of our paper entitled “Fine-scale distribution of malaria mosquitoes biting or resting outside human dwellings in three low-altitude Tanzanian villages”.

With regards to reviewer #3’s main concern (i.e. choice of traps that might not be the best to study outdoor transmission), we have provided further clarification on the rationale behind the choice of Suna and Resting bucket traps, including key references to justify our experimental design in this specific ecological setting.

With regards to the reviewer’s second concern (i.e. pooling the other Anopheles species or keeping them separated), we have discussed this extensively between ourselves, and the large number of field collaborators taking part in the extended project. Given the low contribution to malaria transmission given by non-An. arabiensis and non-An. funestus vectors, we decided not to put too much emphasis on the other Anopheles species. In addition, if we should consider each species separately, the length of the manuscript would increase substantially, and would pull the reader’s attention out of the main focus of our manuscript (i.e. focusing on vectors known to be responsible for malaria transmission in our study area). 

We have now included further clarification with respect to the previous points, by modifying some paragraphs in the data analysis and discussion sections, and by adding mor relevant references.

Finally, we made several minor changes (and typos corrections) to further improve the reading flow. We believe our manuscript has substantially improved.

Please find our detailed point-by-point response to each reviewer’s report. We have highlighted in yellow the specific changes/additions made, in the manuscript main body.

Once again thank you very much for your support and hope to hear from you. 

Yours sincerely

Arnold 

ammbando@ihi.or.tz

Response to reviewer’s comments

Reviewer # 3

Comments 1: The choice of traps might not be the best to study outdoor transmission, as raised by the authors themselves on lines 387-389, because host-seeking traps and humans in the vicinity introduced a bias, An. funestus being highly anthropophilic, this species will avoid the traps to feed on humans. This is particularly obvious in Table 1a in which only 155 specimens of An. funestus s.l. where collected compared to 1556 specimens of An. arabiensis. Is this small number due to the traps that were not appropriate for this species or to its low density? Therefore, the choice of the 2 types of traps should be better explained.

 Response: Thank you very much for this constructive comment. The higher number of Anopheles arabiensis vector abundance over Anopheles funestus reflects the specific ecological setting of our study area. In Kilombero valley previous malaria vector surveillance works caught significant higher number of An. arabiensis as compared to An. funestus mosquitoes (see for example Kaindoa,.et al 2017) [1]. However, about 80% of malaria transmission occurring in the study area are dominated by An. funestus and only 20% was due to An. arabiensis [1]. Apart from the difference in vector composition of these primary vectors, feeding and resting behaviours shown by these two vectors are highly different, with An. funestus showing preference for indoor biting and resting and An. arabiensis outdoor. 

In our study we based the choice of our traps (Suna-traps, and Resting bucket (RBu)) based on the results of several other studies focusing on outdoor sampling in both malaria and non-malaria vectors [2-4]. We have now expanded the method section providing further justification for the choice the traps. Please see lines 159 - 164 in the revised version. Furthermore, we could have used HLC method, however this sampling methods is currently not recommended due to ethical concerns of exposing volunteers to mosquito bites.

Comment 2: Seven mosquito taxa have been collected during the 12 months collection (Table 1b). However, the manuscript is focusing on 2 species only, An. arabiensis and An. funestus. The third category is named "Other Anopheles". The paper would gain in clarity and robustness if the other species were also taken into consideration, especially as the data are already available (Table 1b). When possible, the spatio-temporal analysis should be more specific, including for instance An. zeimanni which was collected in high numbers (n=5607 specimens), 5 times more than An. arabiensis.

 Response: Thank you very much for pointing this out. We agree that splitting the other malaria vectors by each species would be of interest. However, please note that the relative contribution of non- An. arabiensis and non-An. funestus vectors to malaria transmission in this specific ecological setting is low (as we also highlighted in the revised introduction section). Therefore, we deliberately decided not to put too much emphasis on the other Anopheles species, to avoid distracting the reader from the main focus of our work (i.e. outdoor biting and resting of vectors responsible for malaria transmission). Furthermore, please note that if we should expand the analysis to each separate species, the length of our manuscript (and figures and tables) would increase substantially at the cost of losing conciseness 

Additional comment: They are also a good number of minor corrections and comments that have been included into the attached manuscript for its improvement (see attached file). For instance, they are some statements that need further explanation to increase their understanding.

 Response: We really appreciate your additional efforts in providing these editorial suggestions. We have revised the manuscript accordingly. Please find the revised edits highlighted in yellow in the manuscript with track changes. 

References

1. Kaindoa EW, Matowo NS, Ngowo HS, Mkandawile G, Mmbando A, Finda M: Interventions that effectively target Anopheles funestus mosquitoes could significantly improve control of persistent malaria transmission in south-eastern Tanzania. PLoS One 2017, 12.

2. Hiscox A, Otieno B, Kibet A, Mweresa CK, Omusula P, Geier M, Rose A, Mukabana WR, Takken W: Development and optimization of the Suna trap as a tool for mosquito monitoring and control. Malaria journal 2014, 13(1):257.

3. Mburu MM, Zembere K, Hiscox A, Banda J, Phiri KS, Van Den Berg H, Mzilahowa T, Takken W, McCann RS: Assessment of the Suna trap for sampling mosquitoes indoors and outdoors. Malaria journal 2019, 18(1):51.

4. Kreppel KS, Johnson P, Govella N, Pombi M, Maliti D, Ferguson H: Comparative evaluation of the Sticky-Resting-Box-Trap, the standardised resting-bucket-trap and indoor aspiration for sampling malaria vectors. Parasites & vectors 2015, 8(1):462.

---

## [Decision Letter · Decision Letter 2]

11 Dec 2020

PONE-D-20-16651R2

Fine-scale distribution of malaria mosquitoes biting or resting outside human dwellings in three low-altitude Tanzanian villages

PLOS ONE

Dear Dr. Mmbando,

Thank you for submitting your manuscript for review to PLoS ONE. After careful consideration, we feel that your manuscript will likely be suitable for publication if the authors revise it to address additional  points raised by the reviewer.  According to reviewer, there are some specific areas where further improvements would be of substantial benefit to the readers.    

We look forward to receiving your revised manuscript.

Kind regards,

Luzia Helena Carvalho, Ph.D.

Academic Editor

PLOS ONE

Reviewers' comments:

Reviewer's Responses to Questions

**Comments to the Author**

1. If the authors have adequately addressed your comments raised in a previous round of review and you feel that this manuscript is now acceptable for publication, you may indicate that here to bypass the “Comments to the Author” section, enter your conflict of interest statement in the “Confidential to Editor” section, and submit your "Accept" recommendation.

Reviewer #3: (No Response)

2. Is the manuscript technically sound, and do the data support the conclusions?

Reviewer #3: Yes

3. Has the statistical analysis been performed appropriately and rigorously? 

Reviewer #3: Yes

4. Have the authors made all data underlying the findings in their manuscript fully available?

Reviewer #3: Yes

5. Is the manuscript presented in an intelligible fashion and written in standard English?

Reviewer #3: Yes

6. Review Comments to the Author

Reviewer #3: The authors have properly addressed the two main comments raised in the first review of the manuscript.

However, there are stlll some minor revisions that need to be done for improving the manuscript (see below).

Page 2, lines 49-50, write: "in the Anopheles gambiae complex and ...".

Page 5, lines 162-164, write: "The Suna® traps proved to catch significantly higher number of Anopheles species in field conditions, as well as it significantly reduced entry of malaria vectors ...". Write "Anopheles" in italics (line 163).

Page 8, line 254, add "%" after 52.3%.

Page 12, in Table 2, write "Anopheles" in italics in "Other Anopheles species" (top right side of table).

Page 13, line 364, write "as previously described by Mala and Irungu [41]". Line 369, delete the coma between Ref No 42, 43 and (Fig. S1.11).

Page 14, line 386, delete "was" after "partly".

Page 15, lines 407-412, this paragraph is still not logical. High abundance is contradictory with the fact larval stages are washed away during the rainy season. This paragraph still needs improvement and more clarity. Line 428, delete the space after "studies".

Page 20, Ref 35, line 587, write "Anopheles" in italics. Line 597, complete Ref 40 with volume, pages, journal, etc.

Page 21, Ref 55, line 646, write "Anopheles" in italics.

One of the 4 references mentioned in the response to reviewers is missing from the manuscript. It's the one by Mburu et al 2019.

7. PLOS authors have the option to publish the peer review history of their article (what does this mean?). If published, this will include your full peer review and any attached files.

Reviewer #3: **Yes: **Sylvie MANGUIN

---

## [Author Response · Author response to Decision Letter 2]

28 Dec 2020

28th December 2020

Dear Editor, PLOS ONE

Re: Research paper re-submission (Manuscript code: PONE-D-20-16651)

We would like to thank you and the reviewers for constructive comments and for offering us the opportunity to resubmit a revised version of our paper entitled “Fine-scale distribution of malaria mosquitoes biting or resting outside human dwellings in three low-altitude Tanzanian villages”.

Please find the detail explanation of the minor comments from Review #3 listed in point by point below. We also included further clarification with respect to the previous points, by modifying some sentence in the main manuscript, and by adding more relevant references.

Finally, we made several minor changes (and typos corrections) to further improve the reading flow. We believe our manuscript has substantially improved.

Please find our detailed point-by-point response to each reviewer’s report. We have highlighted in yellow the specific changes/additions made, in the manuscript main body.

Once again thank you very much for your support and hope to hear from you. 

Yours sincerely

Arnold 

ammbando@ihi.or.tz

Response to reviewer’s comments

Reviewer # 3

Comments 1: The authors have properly addressed the two main comments raised in the first review of the manuscript.

However, there are stlll some minor revisions that need to be done for improving the manuscript (see below).

 Response: Thank you very much for point out this minor comments. Please find the edits listed below and highlighted in yellow in each respective page number. 

Comment 2: Page 2, lines 49-50, write: "in the Anopheles gambiae complex and ..

 Response: We really appreciate the additional edits. This is now added in the respective page in the main manuscript body. 

Comment 3: Page 5, lines 162-164, write: "The Suna® traps proved to catch significantly higher number of Anopheles species in field conditions, as well as it significantly reduced entry of malaria vectors ...". Write "Anopheles" in italics (line 163).

 Response: We really appreciate the additional edits. This is now italicized in the respective page in the main manuscript body. 

Comment 4: Page 8, line 254, add "%" after 52.3%.

 Response: Thank you so much reviewer for this edit. This is now added in the respective page in the main manuscript body. 

Comment 5: Page 12, in Table 2, write "Anopheles" in italics in "Other Anopheles species" (top right side of table).

 Response: Thank you so much reviewer for this edit. This is now italicized in the respective page in the main manuscript body.

Comment 6: Page 13, line 364, write "as previously described by Mala and Irungu [41]". Line 369, delete the coma between Ref No 42, 43 and (Fig. S1.11).

 Response: Thank you so much reviewer for the edits. This is commas are now removed in the main manuscript body.

Comment 7: Page 14, line 386, delete "was" after "partly"

 Response: Thank you so much reviewer for the edits. This is commas are now removed in the main manuscript body.

Comment 8: Page 15, lines 407-412, this paragraph is still not logical. High abundance is contradictory with the fact larval stages are washed away during the rainy season. This paragraph still needs improvement and more clarity. Line 428, delete the space after "studies".

 Response: Thank you so much reviewer pointing out this. This is caused by heterogeneity of malaria vector densities at small scale and that there may be more factors influencing these densities than we assessed. This line is now added in the main manuscript body, in Page 15, Line 415-416. 

Comment 9: Page 20, Ref 35, line 587, write "Anopheles" in italics. Line 597, complete Ref 40 with volume, pages, journal, etc.

 Response: Thank you for point this out. The references are now edited. 

Comment 10: Page 21, Ref 55, line 646, write "Anopheles" in italics.

Response: This is well noted. Please see it italicized in the main manuscript. 

Comment 11: One of the 4 references mentioned in the response to reviewers is missing from the manuscript. It's the one by Mburu et al 2019.

 Response: Thank you for this suggestion. The reference is now added in the main manuscript, Page 5, line 164

---

## [Decision Letter · Decision Letter 3]

5 Jan 2021

PONE-D-20-16651R3

Fine-scale distribution of malaria mosquitoes biting or resting outside human dwellings in three low-altitude Tanzanian villages

PLOS ONE

Dear Dr. Mmbando,

Thank you for resubmitting your manuscript for review to PLoS ONE. After careful consideration, we feel that your manuscript will likely be suitable for publication if it is revised to address specific queries raised by the reviewer. As quoted by the reviewer, the authors did not properly address relevant topics raised during the peer review process. At this time, we strongly recommend that the authors include/clarify  all topics  raised by the reviewer.

We look forward to receiving your revised manuscript.

Kind regards,

Luzia Helena Carvalho, Ph.D.

Academic Editor

PLOS ONE

Reviewers' comments:

Reviewer's Responses to Questions

**Comments to the Author**

1. If the authors have adequately addressed your comments raised in a previous round of review and you feel that this manuscript is now acceptable for publication, you may indicate that here to bypass the “Comments to the Author” section, enter your conflict of interest statement in the “Confidential to Editor” section, and submit your "Accept" recommendation.

Reviewer #3: (No Response)

2. Is the manuscript technically sound, and do the data support the conclusions?

Reviewer #3: Yes

3. Has the statistical analysis been performed appropriately and rigorously? 

Reviewer #3: Yes

4. Have the authors made all data underlying the findings in their manuscript fully available?

Reviewer #3: Yes

5. Is the manuscript presented in an intelligible fashion and written in standard English?

Reviewer #3: Yes

6. Review Comments to the Author

Reviewer #3: I noticed that some comments have been taken into consideration, while others not.

The previous comments listed and sent to the authors have been made to improve the manuscript. I believe the authors forgot to integrate some of them or misunderstood some of my comments. Another possibility is that they disagreed with some of my comments, in this case an answer is expected to be provided.

I'm listing some comments again hoping this time they will all be taken into consideration.

- Page 2, line 49, add "the" before "Anopheles gambiae complex".

- Page 5, lines 162-164, improve English syntax in writing: "proved ... significantly higher ... in field conditions, ... significantly reduced ... malaria vectors ...".

- Page 8, line 254, delete space before %.

- Page 13, line 364, write "by Mala and Irungu". There are 2 authors only, so Mala et al is not appropriate.

- Page 15, line 418, write "Anopheline", not Anopeline".

- Page 20, lines 612-613, complete Ref 41 by Coosemans & Mouchet with volume, pages, journal, etc.

All references must follow the journal recommendations.

7. PLOS authors have the option to publish the peer review history of their article (what does this mean?). If published, this will include your full peer review and any attached files.

Reviewer #3: No

---

## [Author Response · Author response to Decision Letter 3]

6 Jan 2021

6th January 2021

Dear Editor, PLOS ONE

Re: Research paper re-submission (Manuscript code: PONE-D-20-16651R3)

We would like to thank you and the reviewers for constructive comments and for offering us the opportunity to resubmit a revised version of our paper entitled “Fine-scale distribution of malaria mosquitoes biting or resting outside human dwellings in three low-altitude Tanzanian villages”.

Please find the detail explanation of the minor comments from Review #3 listed in point by point below. We also included further clarification with respect to the previous points, by modifying some sentence in the main manuscript, and by adding more relevant references. We have highlighted in yellow the specific changes/additions made, in the manuscript main body.

Once again thank you very much for your support and so sorry for the comments which were not well addressed during the previous review. We have now addressed all the comments which can be found in the main manuscript body. 

Yours sincerely

Arnold 

ammbando@ihi.or.tz

Response to reviewer’s comments

Reviewer # 3

Comments 1: I noticed that some comments have been taken into consideration, while others not.

The previous comments listed and sent to the authors have been made to improve the manuscript. I believe the authors forgot to integrate some of them or misunderstood some of my comments. Another possibility is that they disagreed with some of my comments, in this case an answer is expected to be provided. I’ m listing some comments again hoping this time they will all be taken into consideration.

Response: Thank you very much for point out additional minor comments. Please find the edits listed below and highlighted in yellow in each respective page number. 

Comment 2: Page 2, line 49, add "the" before "Anopheles gambiae complex

Response: We really appreciate the additional edits. This is now added in the respective page in the main manuscript body. 

Comment 3: Page 5, lines 162-164, improve English syntax in writing: "proved ... significantly higher ... in field conditions, ... significantly

Response: We really appreciate the additional edits. This is now revised in the respective page in the main manuscript body. Please see it in Page 162-164. 

Comment 4: Page 8, line 254, delete space before %.

Response: Thank you so much reviewer for this edit. The space is now removed in the respective page in the main manuscript body. 

Comment 5: Page 13, line 364, write "by Mala and Irungu". There are 2 authors only, so Mala et al is not appropriate.

Response: Thank you so much reviewer for this edit. The reference is now edited accordingly in the main manuscript body.

Comment 6: Page 15, line 418, write "Anopheline", not Anopeline".

Response: Thank you so much reviewer for the edits. This is now edited and can be found in the main manuscript body. 

Comment 7: Page 20, lines 612-613, complete Ref 41 by Coosemans & Mouchet with volume, pages, journal, etc.

Response: Thank you so much reviewer for the edits. The reference number 41 is now edited accordingly in manuscript body.

---

## [Editor Report · Decision Letter 4]

7 Jan 2021

Fine-scale distribution of malaria mosquitoes biting or resting outside human dwellings in three low-altitude Tanzanian villages

PONE-D-20-16651R4

Dear Dr. Mmbando,

We’re pleased to inform you that your manuscript has been judged scientifically suitable for publication and will be formally accepted for publication once it meets all outstanding technical requirements.

Kind regards,

Luzia Helena Carvalho, Ph.D.

Academic Editor

PLOS ONE
---

## [Editor Report · Acceptance letter]

12 Jan 2021

PONE-D-20-16651R4 

Fine-scale distribution of malaria mosquitoes biting or resting outside human dwellings in three low-altitude Tanzanian villages 

Dear Dr. Mmbando:

I'm pleased to inform you that your manuscript has been deemed suitable for publication in PLOS ONE. Congratulations! Your manuscript is now with our production department. 

Kind regards, 

on behalf of

Dr. Luzia Helena Carvalho 

Academic Editor

PLOS ONE